# Recent Advances in Cell-Based Therapies for Ischemic Stroke

**DOI:** 10.3390/ijms21186718

**Published:** 2020-09-14

**Authors:** Satoshi Suda, Chikako Nito, Shoji Yokobori, Yuki Sakamoto, Masataka Nakajima, Kota Sowa, Hirofumi Obinata, Kazuma Sasaki, Sean I. Savitz, Kazumi Kimura

**Affiliations:** 1Department of Neurology, Nippon Medical School, Tokyo 113-8602, Japan; cnito@nms.ac.jp (C.N.); yuki-sakamoto@nms.ac.jp (Y.S.); masa-nakajima@nms.ac.jp (M.N.); 5058-ksowa@nms.ac.jp (K.S.); k-kimura@nms.ac.jp (K.K.); 2Department of Emergency and Critical Care Medicine, Graduate School of Medicine, Nippon Medical School, Tokyo 113-8602, Japan; shoji@nms.ac.jp (S.Y.); haj11570@gmail.com (H.O.); k-sasaki@nms.ac.jp (K.S.); 3Institute for Stroke and Cerebrovascular Disease, UTHealth, Houston, TX 77030, USA; sean.i.savitz@uth.tmc.edu

**Keywords:** angiogenesis, clinical trial, inflammation, neurogenesis, stem cell, stroke

## Abstract

Stroke is the most prevalent cardiovascular disease worldwide, and is still one of the leading causes of death and disability. Stem cell-based therapy is actively being investigated as a new potential treatment for certain neurological disorders, including stroke. Various types of cells, including bone marrow mononuclear cells, bone marrow mesenchymal stem cells, dental pulp stem cells, neural stem cells, inducible pluripotent stem cells, and genetically modified stem cells have been found to improve neurological outcomes in animal models of stroke, and there are some ongoing clinical trials assessing their efficacy in humans. In this review, we aim to summarize the recent advances in cell-based therapies to treat stroke.

## 1. Introduction

Although early interventions to treat damage caused by reperfusion such as intravenous thrombolysis and endovascular revascularization have shown significant benefits in stroke patients, stroke remains a leading cause of long-term disability worldwide. Therefore, stroke is associated with socioeconomic problems resulting from factors such as increased family burden and medical costs. Experimental laboratory results of stem cell-based therapy using different cell types have been promising, and some clinical trials are beginning to prove the safety and efficacy of this intervention [1,2,3,4,5,6,7,8]. In this review, we aim to summarize the studies of bone marrow mononuclear cells (MNCs), bone marrow mesenchymal stem (BMSCs), dental pulp stem cells (DPSCs), neural stem cells (NSCs), induced pluripotent stem cells (iPSCs), and genetically modified stem cells used for stem cell therapy, including their mechanisms of action and the beneficial effects following stroke in animal models and human studies.

## 2. Bone Marrow Mononuclear Cells

Bone marrow MNCs include a population of monocytes, lymphocytes, mesenchymal and hematopoietic stem cells, and hematopoietic and endothelial progenitor cells [9]. Stem cells such as BMSCs and iPSCs require a period of cell culture before transplantation, whereas MNCs can be collected autologously just prior to administration, which could be advantageous in acute clinical care settings compared with other cell sources.

### 2.1. Protective Mechanisms of Bone Marrow Mononuclear Cells against Stroke

The experimental rationale for the use of MNCs in stroke therapy includes a number of mechanisms of action, such as the modulation of local and systemic inflammation, promotion of angiogenesis and endogenous neurogenesis, differentiation into cell types that facilitate cellular repair processes, and secretion of neurotrophic factors from the acute phase to the chronic phase after stroke [10,11,12] (Figure 1) (Table 1). The main mechanisms are considered to be angiogenesis and a reduction of endothelial damage.

MNCs increase vascular density and blood flow in patients with various ischemic disorders, such as cardiovascular disease, peripheral arterial disease, and diabetic foot pathologies [13,14,15,16,17,18]. Initial studies indicated that differentiation of hematopoietic stem cells into endothelial cells (ECs) was the major contributory mechanism of recovery. Subpopulations of MNCs, such as CD34^+^/M-cadherin^+^ cells, can promote angiogenesis and arteriogenesis by differentiating into smooth muscle cells (SMCs) and ECs in ischemic hindlimbs of rodents [19,20]. Wang et al. reported that transplanted MNCs had the capacity to differentiate into SMCs and ECs after permanent middle cerebral artery occlusion (MCAO) in rats [20,21]. These findings illustrate that MNCs are involved in angiogenesis and arteriogenesis, which may contribute to improved blood flow restoration in ischemic tissue. In a mouse model of bilateral common carotid artery stenosis, MNC treatment induced an increase in cerebral blood flow (CBF) through the upregulation of Ser1177 phosphorylation and increased endothelial nitric oxide synthase levels beginning in the early phase after stoke, as well as subsequent restoration of endogenous responses in the later phase, including angiogenesis [22]. However, in that study, the authors observed benefits from MNC treatment despite a lack of evidence of direct structural incorporation of donor MNCs into ECs. Also, subsequent studies showed that transdifferentiation of grafted MNCs into an EC phenotype was only rarely observed (<1%) [23,24]. Very recently, Taguchi et al. demonstrated that MNCs accelerate vascular endothelial growth factor (VEGF) uptake into ECs and suppress autophagy through gap junction-mediated cell–cell interactions [25]. These findings provided evidence of a novel mechanism through which a prominent gap junction-mediated signaling pathway activates angiogenesis and supports the survival of injured ECs after ischemia.

### 2.2. Clinical Studies of Bone Marrow Mononuclear Cells Efficacy Following a Stroke

Savitz et al. previously reported that intravenous infusion of autologous MNCs within 24–72 h after a stroke might be effective compared with a control group of age- and National Institutes of Health Stroke Scale (NIHSS) score-matched historical stroke patients [31]. In addition, Taguchi et al. reported that intravenous infusion of autologous MNCs with 7–10 days of the onset of stroke was a safe and feasible therapy that led to improved functional recovery and increased CBF and metabolism in stroke patients with severe ischemia (NIHSS score ≥ 10 on day 7 post-stroke) [32]. On the other hand, a phase II, multicenter, randomized clinical trial demonstrated that intravenous infusion of autologous MNCs at a median of 18.5 days after stroke onset was safe, but they observed no beneficial effect of treatment on stroke outcomes [33]. These clinical results indicate that earlier transplantation of MNCs soon after stroke onset may be more efficacious in promoting recovery. Since MNCs offer the particular advantage of acute and autologous transplantability, age, sex, and underlying comorbidities might influence both the patients’ susceptibility to and the functionality of MNC grafts, though more research is needed to identify the impact of these factors on the efficacy of MNC transplantation for stroke treatment [34].

## 3. Bone Marrow Mesenchymal Stem Cells

BMSCs have self-renewal potential. These cells express markers of mesenchymal cells or ECs (CD105, CD73, and CD90), as well as adhesion molecules (CD106, CD166, and CD29), but not hematopoietic stem cell markers (CD11, CD14, CD34, CD45, CD79, CD19, and human leukocyte antigen DR isotype [HLA-DR]) [35,36]. BMSCs are easily cultured in vitro, have weak immunogenicity and a good safety profile, and have been considered to be ideal seed cells in the treatment of ischemic stroke [37].

### 3.1. Protective Mechanisms of Bone Marrow Mesenchymal Stem Cells against Stroke

BMSCs have been widely investigated in experimental stroke models (Table 2). Studies have shown that BMSCs have the ability to differentiate into neuronal cells in vitro in coculture or differentiation medium. BMSCs can also influence the microenvironment at an injury site through the secretion of anti-inflammatory factors or by decreasing interleukin 1 beta (IL-1β), interleukin 6 (IL-6), and tumor necrosis factor alpha (TNF-α) levels. They may also induce the secretion of antiapoptotic molecules and trophic factors that promote angiogenesis, immunomodulation, including inhibition of T-cell proliferation, promotion of regulatory T cell (Treg) function, and diminishment of IL-23/IL-17 expression, and increased axonal growth [37,38,39,40,41].

### 3.2. Efficacy and Safety of Bone Marrow Mesenchymal Stem Cells in Clinical Trials

In a prospective, randomized study of 30 patients with severe stroke, Bang et al. showed that BMSCs improved the modified Rankin scale (mRS) and Barthel index scores up to a year after stroke [1]. Serial evaluations showed no adverse cell-related, serological, or imaging-defined effects. Similarly, Honmou et al. reported that intravenous infusion of autologous BMSCs expanded in human serum into 12 subjects 36–133 days post-stroke did not result in any significant adverse events [4].

The SanBio study assessed the use of SB623 cells, which are allogenic modified BMSCs transiently transfected with the human Notch-1 intracellular domain, in a two-year, single-arm, open-label, uncontrolled study of 18 patients in a stable chronic phase after stroke (6–60 months post-stroke) [48]. SB623 cells were injected, stereotactically, into the infarct area and the subjects were followed for 24 months. Significant improvements were observed based on the European Stroke Scale score, the NIHSS score, Fugl-Meyer total score, and Fugl-Meyer motor score; however, there was no statistically significant improvement of the subjects’ mRS scores. The efficacy of the treatment plateaued within one year, without decreasing thereafter [48]. Recently, Savitz et al. conducted a randomized, sham controlled, phase II trial in which subjects received intracarotid delivery of aldehyde dehydrogenase (ALDH)–bright cells (13 and 19 days poststroke) isolated from the bone marrow of patients who had recently experienced an ischemic middle cerebral artery stroke [6]. Although the study provided a framework for conducting randomized, sham-controlled trials involving intracarotid administration of autologous, sorted BMSCs in patients with a recent stroke, there were no significant differences between the groups based on any of the efficacy measures (mRS at 90 days, as wells as disability assessed by Barthel index scores, quality of life based on European Quality of Life- 5 Dimension (EQ-5D) scores, and rehabilitation utilization at one year).

Unlike the studies described above, the RAINBOW trial was a first-in-human trial to use superparamagnetic iron oxide-labelled BMSCs to treat patients following a subacute ischemic stroke. Preliminary results indicated that intracerebral transplantation of autologous BMSCs was safe and well tolerated. Moreover, it is expected that bio-imaging techniques could help clarify the therapeutic mechanisms of action [7,49]. Taken together, these studies support the safety of BMSCs or BM-derived cells for transplantation in patients with ischemic stroke, although the therapeutic efficacy remains controversial. Further studies are needed to identify the optimal transplantation protocol for routine clinical applications, including the concentration of cells, the timing, the route of administration, patient selection criteria (age, stroke subtype, and location of damage), and combination therapies.

## 4. Dental Pulp Stem Cells

Human DPSCs are the neural crest-derived stem cells residing within the perivascular niches of the dental pulp [50] and are an attractive cell source because they can be easily obtained as medical waste without ethical or logistical complications [51]. As human DPSCs originate from the neuronal lineage, they are considered a particularly promising strategy for treatment of severe neurological disorders. DPSC isolation from extracted teeth is easily performed and less invasive than the isolation of BMSCs. DPSCs express a variety of cell surface markers similar to BMSCs, such as CD29, CD90, and CD105, but they do not express hematopoietic markers, such as CD34 and CD45 [52]. Human DPSCs also exhibit approximately three times higher proliferation in vitro than human BMSCs [53]; they are multipotent and can differentiate into muscle, cartilage, bone, and other cell types [54]. In addition, DPSCs have also been shown to exert potent immune-modulatory properties through the inhibition of activated T-cell responses [51]. The immunosuppressive properties of DPSCs make them attractive for use in allogeneic transplantation. The neurotrophic factors secreted by DPSCs are involved in processes mediated by nerve growth factor (NGF), neurotrophin-3, brain-derived neurotrophic factor (BDNF), glial cell-derived neurotropic factor (GDNF), and vascular endothelial growth factor (VEGF); these molecules promote neuronal survival, cellular proliferation, differentiation, and migration [55]. DPSCs exhibit properties of both neural stem cells and mesenchymal stem cells (MSCs), with reported utility for the treatment of cerebral ischemia [56,57,58]. Several studies have shown that DPSC transplantation is associated with neuroprotective effects and enhances functional recovery after cerebral ischemia in vivo (Table 1) [59,60,61,62,63,64] (Table 3); however, more basic science studies are needed to confirm a safe and efficacious method for DPSC transplantation during the acute phase after ischemic stroke.

### 4.1. Protective Effects of Dental Pulp Stem Cells after Ischemic Stroke In Vivo

Multipotent stem cells, such as DPSCs, have limited differentiation potential, which makes them potentially safer for clinical use, with no authenticated risk of tumor formation [57,65,66]. Recent studies have also shown that engrafted human DPSCs can survive in damaged central nervous system (CNS) tissue and exert immunomodulatory effects by upregulating anti-inflammatory cytokines and attenuating pro-inflammatory cytokines secreted from activated microglia and macrophages in murine xenogenic transplantation models without the use of immunosuppressive drugs [67,68,69]. The mechanisms of action of exogenous DPSC treatment in various in vivo experiments have been attributed to their paracrine effects [70], which are mediated by the release of secreted factors, cytokines, chemokines, and growth and trophic factors, including the stromal cell-derived factor-1, NGF, BDNF, GDNF, and VEGF [71,72]. The autocrine and paracrine effects of DPSCs have been proposed to be responsible for improving the microenvironment of host cells and for enhancing endogenous restorative processes following injury or disease. Indeed, stem cell transplantation is also regarded as a cell-based cytokine therapy [73]. For example, IL-10, which is produced by regulatory T lymphocytes, reduces pro-inflammatory cytokine production and the neurotoxicity associated with TNF-α and interferon gamma (IFN-γ) signaling in ischemic brain injury [74]. DPSC transplantation suppresses TNF-α and IL-1β levels, which are markers of systemic inflammation in both the brain and serum, 72 h after reperfusion injury [69]. Similarly, VEGF expressed in macrophages, neurons, glia, and vascular ECs promotes angiogenesis in the ischemic brain during tissue repair [75].

A previous study has shown that human DPSCs could potentially differentiate into functional neural progenitor cells or neurons, which can integrate with other brain tissues [76]. Other studies showed that transplanted DPSCs were able to migrate to the boundaries of ischemic areas, differentiate into neuron- and astrocyte-like cells in the rat brain [61], and express neuronal and NSC markers such as βIII tubulin, doublecortin, nestin, and neurofilament [64]. However, very few DPSCs were able to survive in the ischemic brain, and most migrated to the peri-infarct areas where they differentiated mostly into astrocytes rather than neurons. Thus, the limited survival, differentiation, and integration of DPSC-derived cells into the ischemic lesion implies that functional improvements are more likely to be mediated through bystander effects rather than as a result of cell replacement and differentiation. Although the precise neuroprotective mechanisms associated with transplanted DPSCs remain unclear, they are likely related, at least in part, to anti-inflammatory and angiogenic factors.

The optimal route of transplanted stem cells has been investigated, with studies showing that intravascular delivery was the predominant route, followed by intracranial or intraventricular routes [77]. Intra-arterial injection, which facilitates higher levels of cell engraftment in the target organ, may increase the risk for the development of a microembolism after cell transplantation [78]. This means that the incidence of MSC-induced vascular obstructions and stroke after intra-carotid injection is closely related to the size of the cells [79], and several studies have reported that the intravenous administration of human DPSCs decreased ischemic damage and promoted functional improvement in a rodent model of focal cerebral ischemia (Table 1). Intravenous administration of DPSCs can reduce the risk of arterial microembolism since the cell sizes are similar to those of MSC populations; however it is important to consider the filtering effect as the cells pass through the lungs after systemic delivery.

### 4.2. Clinical Studies of Dental Pulp Stem Cells Administration after a Stroke

Intravenous administration of DPSCs is less invasive than intracerebral or intraventricular transplantation, thereby reducing trauma to stroke patients. Moreover, systemic delivery of the cells allows for easier transmigration through the blood brain barrier (BBB) to the ischemic hemisphere, and it is an effective and safe procedure for acute stroke patients [4,47,80]. Clinical investigations involving intravenous injection of human allogenic DPSCs have already begun in Japan. JTR-161 is a first-in-human, randomized, double-blind, placebo-controlled, multicenter clinical trial to evaluate the safety and efficacy of DPSC administration in patients with ischemic stroke. This study was designed in accordance with good manufacturing practice by JCR Pharmaceuticals Co., Ltd. and consists of three cohorts. The DPSCs used in the study are diluted in 100 mL of saline for a single intravenous administration after stroke onset. Patients are recruited from 29 stroke centers in Japan between December 2018 and July 2021.

## 5. Neural Stem Cells

In 1962, the possibility of neural proliferation was described in adult rats by Altman [81]. In 1992, Reynolds, Weiss, and colleagues isolated NSCs and propagated them in the presence of epidermal growth factor (EGF) for the first time to give rise to large cellular spheres that they termed “neurospheres” [82,83]. Neurons and glial cells are derived from common immature NSCs, which are defined as self-renewing and multipotent cells that can differentiate into neurons, astrocytes, and oligodendrocytes. NSCs have been found to exist not only in the developing brain but also in the mature mammalian brain. NSCs exist in at least two regions of the adult brain—the subventricular zone (SVZ) of the lateral ventricle and the subgranular zone (SGZ) of the hippocampus. NSCs in the SVZ mainly migrate along the rostral migratory stream to the olfactory bulb, whereas NSCs in the SGZ migrate to the granule cell layer where they finally differentiate into various neural cells and integrate into neuronal networks [84]. In addition, the human brain may contain another stem cell pool in the deep ventral region of the prefrontal cortex due to the highly developed prefrontal lobe [85].

NSCs can be derived from several sources, including embryonic stem cells and fetal tissue. Two main processes regulate the differentiation of NSCs, like self-regulation and exogenous signal regulation [86]. Self-regulation is controlled in the developmental period by several intrinsic and transcription factors. Activation of Notch-mediated signaling initiates the proliferation of NSCs, thereby increasing their numbers. In addition, the self-regulation of NSCs is controlled by other transcription factors or pathways, such as the Wnt and the Sonic hedgehog signaling pathways [87]. Regulation by exogenous signaling is also important for the differentiation of developing NSCs; NSCs originating from the same source can differentiate into different types of cells in the CNS based on the local microenvironment in brain tissue. This exogenous signal regulation of NSC proliferation, differentiation, and self-renewal is also mediated by several cytokines and neurological growth factors, including fibroblast growth factor (FGF) [88], EGF [89], BDNF [90], and PDGF [91] (Table 4).

### 5.1. Experimental Studies Investigating Neural Stem Cells Transplantation to Treat Stroke

The discovery of NSCs provided a promising new therapy for the treatment of a variety of neurological diseases and injuries, such as Parkinson’s disease, Huntington’s disease, Alzheimer’s disease, multiple sclerosis, amyotrophic lateral sclerosis, spinal cord injury, traumatic brain injury, and stroke, all of which are characterized by the failure of endogenous repair mechanisms in the CNS to restore damaged tissue and rescue lost functions [92,93]. As shown in Table 2, many studies have assessed NSC transplantation to treat ischemic brain injury in animal models. The initial hypotheses were based on the assumption that NSCs would replace lost neurons and restore connections in neuronal circuitry [94,95,96,97,98,99,100,101,102,103,104,105,106]. Indeed, analysis of graft-derived neuronal cells using immuno-electron microscopy and electrophysiological recording demonstrated increased connectivity, showing that NSCs could develop into functioning neurons to contribute to improved recovery from stroke and brain injury in rats [107,108]. Although transplanted NSCs are expected to differentiate into various neural cells rather than other stem cells, it remains unclear to what extent this contributes to functional recovery. The current understanding is that transplanted NSCs likely prevent neuronal apoptosis, exert immunomodulatory effects inside and outside the brain, increase endogenous neuronal regeneration and angiogenesis, and inhibit glial scar formation mainly via paracrine and autocrine secretion of various neurotrophic factors rather than differentiation [81,82,83,84,85,86,87,88,89,90,91,92,93,94,95,96,97,98,99,100,101,102,103,104,105,106,109,110] (Table 2).

### 5.2. Clinical Studies Investigating Neural Stem Cells Transplantation to Treat Stroke

NSC transplantation has been performed mainly as a treatment during the chronic phase post-stroke, which has as few therapeutic options as treatment during the acute phase. In the early 2000s, many studies investigated the use of NSCs for treatment during the chronic phase; however, there have been few completed clinical trials of cell-based therapy, and those that exists included small cohorts of patients with chronic stroke [111].

The PISCES 1 trial (Pilot Investigation of Stem Cells in Stroke; ClinicalTrials.gov, number NCT01151124) was a first-in-human study using neuronal progenitor cells (NPCs) derived from the human fetal cortex to treat patients in the chronic phase after damage to the basal ganglia. This trial was an open-label, single-site, dose-escalation study [112]. The cells were modified with the c-Myc gene to enable chemical control of proliferation (CTX cells). CTX cells produce multiple growth factors and cytokines that promote angiogenesis and neurogenesis, and reduce inflammation [112,113]. In this study, 11 patients were enrolled from 6–60 months after stroke onset. CTX cells were transplanted stereotactically through a single injection into the ipsilateral putamen. During this trial, no immunological or cell-related adverse events were reported. The patients who received the CTX cells experienced improvements in their NIHSS and Ashworth Spasticity Scale scores, and neurological improvements were seen within one month from the procedure, which were maintained at the two-year follow-up. Moreover, three patients who were followed up for over two years experienced improvement of their mRS scores by 1 point. In addition, hyperintensity around the implantation tracts was seen in five patients based on fluid-attenuated inversion recovery MRI, but this did not correlate with clinical improvements. The study showed the safety of CTX cell transplantation, which could improve neurological function in patients in the chronic phase post-stroke [112].

The PISCES 2 study (ClinicalTrials.gov, number NCT02117635) was a prospective, multicenter, single-arm, open-label study in adults over 40 years of age who experienced significant upper limb motor deficits 2–13 months (median: 7 months) after ischemic stroke [114]. Twenty-three patients received transplantations of 20 million CTX cells into the ipsilateral putamen. No safety concerns related to the cells were reported at one year. One patient demonstrated improved Action Research Arm Test performance at 3 months, and three patients improved after 6–12 months. Seven patients had improved mRS scores by at least 1 point. In total, 15 patients showed improvement on one or more of the clinical scales [114].

Subsequently, the PISCES-3 study (ClinicalTrials.gov, number NCT03629275) was a randomized, placebo-controlled, Phase IIb clinical trial [113]. This trial enrolled about 130 patients 6–24 months after a stroke from 40 centers across the United States who had moderate to moderately severe functional disability (with a baseline mRS score of 3–4). The patients were randomized using a 1:1 ratio (CTX implantation: sham surgery control), and received stereotactic implantations of 20 million CTX cells into the putamen ipsilateral to the cerebral infarct. The sham surgery group received a partial thickness burr-hole without insertion of the cannula or penetration of the dura. The primary outcome was an improvement of mRS scores six months after the procedure, with further monitoring up to 12 months.

## 6. Induced Pluripotent Stem Cells

Mice-induced pluripotent stem (iPS) cells were first established by Yamanaka et al. with introducing four transcription factors (Oct3/4, Klf4, Sox2, and c-myc) into mouse fibroblasts [115]. iPSCs can be an ideal source of cells for cell therapy, as they can be reprogrammed from somatic or blood cells, theoretically from any individual, and have limitless self-renewal capacity and pluripotency, allowing them to differentiate into all cell types derived from the three germ layers (endoderm, mesoderm, and ectoderm) [115,116]. These advantages are not common to other stem cells such as MSCs or NSCs, and the research and application of iPSC products provoke less ethical controversy than embryonic stem cells. Nonetheless, there are still several hurdles to overcome.

### 6.1. Applying Induced Pluripotent Stem Cells in Animal Models of Stroke

To date, the application of iPSCs in cell therapy for stroke has been solely tested in animal stroke models. The earliest studies assessed the effects of transplanting undifferentiated iPSCs into the ischemic brain or the subdural space adjacent to the site of infarction in rat or mouse MCAO models [117,118,119,120]. The results were not consistent in terms of treatment effects, as some observed improvements [117,118] while others did not [119]. Not surprisingly, transplanted iPSCs were associated with tumorigenicity [119,120], which was more evident when the cells were transplanted into ischemic lesions of pathological brain tissue than into healthy brains [120].

Thereafter, differentiated iPSCs or iPSC derivatives were used in animal models [121,122,123,124,125,126,127]. In almost all experimental studies, iPSCs were differentiated into cells that shared characteristics common to immature neuronal cells; these became known as NSCs or NPCs. These NSCs and NPCs were transplanted into ischemic brains, resulting in beneficial treatment effects, such as reduced infarct volume or symptom improvement, and no tumor formation was reported when differentiated iPSCs were used. Transplanted cells could survive for a certain period of time, allowing them to migrate toward the peri-infarct zone and further differentiate into neurons or astrocytes. Most studies concluded that transplanted cells contributed to improvements mainly via secreting trophic factors or immunomodulators to reduce inflammation, support residual tissue and enhance regeneration, or by forming connections with host neural networks, resulting in reduced brain atrophy, rather than neuronal replacement. However, Tornero et al. reported that transplanted iPSC-derived neuroepithelial-like stem cells differentiated into neurons with a cortical phenotype and developed functional synapses with host neurons, showing replacement was possible [128,129].

Recent studies have focused on expanding applications of iPSCs in animal models of aging [130], and tried to determine the optimal timing for transplantation in terms of differentiation states [131,132] and the effect of pathological conditions on transplanted cell viability and differentiation [133]. These studies showed that iPSC-derived cells were also effective in reducing the damage caused by aging or reactive gliosis, and completely differentiated cells could result in decreased viability after transplantation. Moreover, iPSC-derived NPCs were tested in models with larger animals, such as the porcine stroke model, revealing a reduction in inflammation and amelioration of stroke-associated symptoms [134,135].

### 6.2. Merits and Hurdles of Using Induced Pluripotent Stem Cells for Cell Therapy to Treat Stroke

Some reviews have summarized the applications of iPSCs in cell therapies [136]. Most previous attempts aimed to use iPSCs after differentiation into specific cell types, mainly NSCs. Therefore, theoretically, the unique mechanisms through which iPSCs protect against damage compared to other stem cells are unknown. Inherent properties of iPSCs, such as limitless potential for self-renewal and pluripotency could be a huge advantage. For example, iPSCs may be associated with a lower rejection rate following transplantation [137], and genetically engineered iPSCs can be enriched to enhance secretion of specific hormones. In addition, various stages of differentiated cells derived from the same iPSC line could be used to target phase post-stroke; immature iPSC-derived NSCs can suppress inflammation in the acute phase and mature neuroepithelial-like cells could be used as cell replacement in the chronic phase. The therapeutic time window may not be long in the acute phase, so being able to tailor treatment for stroke patients would be beneficial. However, some inherent properties of iPSCs may need to be overcome. Limitless self-renewal is associated with tumorigenesis, and the pluripotency of these cells means there is a need to identify the optimal cell type or differentiation state. Detecting and excluding undifferentiated iPSCs, evaluating the tumorigenicity of the differentiated cells, and developing differentiation methods with xeno-free, chemically defined media is critical. While iPSCs are a promising source for cell therapy, there are many hurdles to be overcome.

## 7. Gene Modifications in Stem Cell Therapy to Repair Stroke-Induced Damage

Stem cell-based gene therapy is a promising and effective therapeutic strategy for stroke. Stem cells act as gene delivery vehicles while also secreting various neurotrophic factors. Transplantation of gene-modified stem cells overexpressing various growth factors or cytokines, such as BDNF, GDNF, NGF, VEGF, hepatocyte growth factor (HGF), placental growth factor (PIGF), angiopoietin-1 (ANG-1), erythropoietin (EPO), IL-10, and Noggin has been demonstrated to significantly promote functional recovery compared to stem cells alone in experimental stroke models. 

### 7.1. Brain-Derived Neurotrophic Factor

BDNF gene-transfected MSCs using an adeno-associated virus (AAV) vector can significantly reduce infarct volumes and improve motor function compared with MSC monotherapy in the rat transient MCAO model [138]. BDNF promotes the survival and differentiation of neuronal tissue by upregulating B-cell lymphoma 2 (Bcl-2) and downregulating Bcl-2 associated X protein (Bax) within the ischemic penumbra [139]. Moreover, BDNF can promote the activation of astrocytes [140], and astrocyte-derived exosomes promote axonal elongation and functional recovery after the subacute phase of stroke [141].

### 7.2. Glial-Derived Neurotrophic Factor

GDNF gene-transfected MSCs using an AAV vector have been shown to significantly reduce infarct volumes and improve motor function in the rat transient MCAO model [138]. GDNF has an important role in neuronal survival through binding to the GFRα1 (GDNF family receptor alpha-1) receptor and activating the receptor tyrosine kinase Ret [142].

### 7.3. Nerve Growth Factor

NGF gene-transfected MSCs using an AAV vector have been shown to promote neural cell survival and improve neurological deficits caused by ischemic stroke in the rat transient MCAO model [143]. NGF produced by MSCs is involved in early neurogenesis and the generation of certain neuropeptides [144].

### 7.4. Vascular Endothelial Growth Factor

Neural stem cells (NSCs) forced to overexpress VEGF using lipofection resulted in functional recovery in the mouse intracerebral hemorrhage (ICH) model [145]. Overexpressing VEGF provided differentiation and survival of grafted human NSCs and renewed angiogenesis in the host brain [145].

### 7.5. Hepatocyte Growth Factor

HGF gene-transfected MSCs using a herpes simplex virus type-1 vector significantly reduced infarct volumes and improved motor function in the rat transient MCAO model [146]. DPSCs overexpressing HGF using an AAV vector also enhanced therapeutic effects of DPSCs in the rat transient MCAO model [63]. HGF attenuates BBB destruction by protecting tight junction proteins such as Zo-1 and occludin [63].

### 7.6. Placenta Growth Factor

PIGF gene-transfected MSCs using an AAV vector significantly reduced infarct volumes and improved motor function in the rat permanent MCAO model [147]. PIGF contributes to angiogenesis after cerebral ischemia as a member of the VEGF family [147].

### 7.7. Angiopoietin-1

ANG-1 gene-transfected MSCs using an AAV vector resulted in structural–functional recovery through improved vascular formation and maturation in the rat permanent MCAO model [148] by contributing to post-stroke angiogenesis and maintaining BBB integrity [149].

### 7.8. Erythropoietin

EPO gene-transfected MSCs using a lentivirus vector promoted neural cell survival and improved neurological deficits caused by ischemic stroke in the rat transient MCAO model [150]. EPO is associated with neurotrophic, anti-oxidant, anti-apoptotic, and anti-inflammatory effects in focal brain ischemia [151].

### 7.9. Interleukin-10

MSCs overexpressing IL-10 using an AAV vector significantly reduced infarct volumes and improved motor function in the rat transient MCAO model [43]. IL-10 plays a neuroprotective and vasculoprotective role in cerebrovascular disorders by attenuating pro-inflammatory signals and upregulating anti-apoptotic proteins, such as Bcl-2 and Bcl-extra-large (Bcl-xL) [152,153]. Moreover, overexpression of IL-10 suppressed neuronal degeneration and improved survival of engrafted MSCs in the ischemic hemisphere [43].

### 7.10. Noggin

Noggin gene-transfected MSCs using an AAV vector significantly improved neurological function in the rat transient MCAO model [154]. Noggin, an antagonist of bone morphogenetic protein (BMP), promotes the differentiation of MSCs into neurons [154]. Moreover, noggin suppressed ischemia-induced apoptosis and inflammation through the protein kinase B/glycogen synthase kinase 3 beta (Akt/GSK3β) and toll-like receptor 4/myeloid differentiation primary response 88 (TLR4/MyD88) pathways in the rat MCAO model [143]. In addition, the co-transfection of Noggin and NGF or BDNF in MSCs induces synergistic effects resulting in improved neurological outcomes and neural remodeling compared to singly transfected (NGF or BDNF) BMSCs [143,154].

## 8. Perspectives

Stem cells have become attractive candidates for cell therapy in stroke treatment, though so far, no ideal therapeutic interventions are available. Although stem cells are associated with various beneficial effects, including neuroprotection, reduced inflammatory and immune responses, and increased angiogenesis and neurogenesis, the main mechanism through which each cell type is able to attenuate the damage caused by a stroke is yet to be clarified. Clinical challenges may include complicating factors, such as the effect of age, co-morbidities, stroke subtype, and stroke severity, all of which can affect the efficacy of cell therapy. It will take coordinated basic science and clinical collaborations before cell therapy can become a standard stroke treatment. In the future, a combination of stem cell and gene therapy will play an important role in experimental models and clinical treatments.

## Figures and Tables

**Figure 1 ijms-21-06718-f001:**
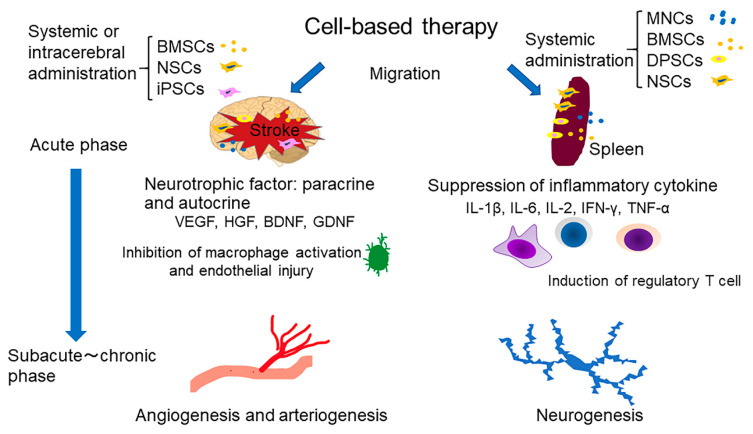
Overview of proposed mechanisms of cell-based stroke therapies. Engrafted therapeutic cells exert neuro- and vaso-protective effects through secretion of various growth factors and systemic inflammation modulation. MNCs, marrow mononuclear cells, BMSCs, bone marrow mesenchymal stem cells; DPSCs, dental pulp stem cells; NSCs, neural stem cells; iPSCs, induced pluripotent stem cells; VEGF, vascular endothelial growth factor; HGF, hepatocyte growth factor; BDNF, brain-derived neurotrophic factor; GDNF, glial cell-derived neurotrophic factor; IL-1β, interleukin 1 beta; IL-6, interleukin 6; IL-2, interleukin 2; IFN-γ, interferon gamma; TNF-α, tumor necrosis factor alpha.

**Table 1 ijms-21-06718-t001:** Experimental study for the bone marrow mononuclear cells (MNCs) transplantation into ischemic brain injury in animal model.

Authors, Year	Cell Type	Number of Cells	Animal Model	Delivery Method	Delivery Timing	Results	Reference
Okinaka, Y. et al. 2019	Human clot-free MNCs	1 × 10^5^	Mice permanent MCAO	Intravenous	48 h post-ischemia induction	Brain atrophy ↓	[26]
Yang, B. et al. 2017	Rat MNCs	1 × 10^7^	Rat embolic stroke model with recombinant tissue plasminogen activator	Intravenous (femoral vein)	3 h post-ischemia induction	Infarct volume → Hemorrhage transformation ↓ BBB permeability ↓ Inflammation modulation	[27]
Li, Y. et al. 2016	MNCs from 5-fluorouracil pre-treated rats	1 × 10^7^	Rat MCAO (120 min)	Intravenous (tail vein)	24 h post-ischemia induction	Infarct volume ↓ Neurological outcome ↑ growth factors ↑	[28]
Suda, S. et al. 2015	Rat MNCs	1 × 10^7^	Rat ICH model	Intravenous (tail vein)	24 h post-ICH induction	Brain edema↓ Brain atrophy↓ Cognitive functional recovery ↑Inflammation modulation Angiogenesis ↑	[10]
Yang, B. et al. 2013	Rat MNCs	1 × 10^7^	Rat MCAO (90 min)	Intravenous and intra-arterial	24h post-ischemia induction	Neurological outcome ↑ Inflammation modulation Neurogenesis ↑	[29]
Nakano-Doi, A. et al. 2010	Mice MNCs	1 × 10^6^	Mice permanent MCAO	Intravenous (tail vein)	24h post-ischemia	Neurological outcome ↑ Cerebral blood flow ↑ Endothelial proliferation ↑ Proliferation of neural stem/progenitor cells ↑	[30]

ICH, intracerebral hemorrhage; MCAO, middle cerebral artery occlusion. “↑”(means increase), “↓” (means decrease), and “→”(means no change).

**Table 2 ijms-21-06718-t002:** Experimental study for the bone marrow mesenchymal stem cell (BMSCs) transplantation into ischemic brain injury in animal model.

Authors, Year	Cell Type	Number of Cells	Animal Model	Delivery Method	Delivery Timing	Results	Reference
Tobin, M.K. et al. 2020	Interferon-γ-activated BMSCs	5 × 10^6^/kg	Rat MCAO (90 min)	Intravenous (retro-orbital sinus)	4.5 h post-ischemia induction	Infarct volume ↓ Neurological outcome ↑ Cerebral blood flow ↑ Oligodendrogenesis ↑ Inflammation modulation	[42]
Nakajima, M. et al. 2017	Human BMSCs Interleukin-10-transfected BMSCs	1 × 10^6^	Rat MCAO (90 min)	Intravenous	0 or 3 h after ischemia reperfusion	Infarct volume ↓ Neurological outcome↑ Inflammation modulation	[43]
Kawabori, M. et al. 2016	Rat BMSCs	1 × 10^5^ or 1 × 10^6^	Rat permanent MCAO	Ipsilateral striatum	1 or 4 weeks post-ischemia induction	Neurological outcome ↑ Differentiation of MSCs	[44]
Toyoshima, A. et al. 2015	Rat BMSCs	1 × 10^6^	Rat MCAO (90 min)	Intra-arterial	1, 6, 24, 48 h after ischemia induction	Infarct volume ↓ Neurological outcome ↑ neurotrophic factor ↑	[45]
Nakazaki, M. et al. 2015	Rat BMSCs	1 × 10^7^	Spontaneously hypertensive rat (stroke-prone) (SHRSP) model	Intravenous	21 weeks of age	Disruption of blood brain barrier ↓ Brain atrophy ↓ Amyloid β accumulation ↓ Cognitive functional recovery ↑	[46]
Wang, L.Q. et al. 2014	Rat BMSCs	1 × 10^4~7^	Rat permanent MCAO	Intravenous (tail vein)	3 and 24 h and 7 days post-ischemia induction	Infarct volume ↓ Neurological outcome ↑ Inflammation modulation	[47]

MCAO, middle cerebral artery occlusion. “↑” (means increase) and “↓” (means decrease).

**Table 3 ijms-21-06718-t003:** Experimental study for the dental pulp stem cells (DPSCs) transplantation into ischemic brain injury in animal model.

Authors, Year	Cell Type	Number of Cells	Animal Model	Delivery Method	Delivery Timing	Results	Reference
Leong, W.K. et al. 2012	Human DPSCs	6 × 10^5^	Rat MCAO (2 h)	Intracerebral (striatum and cortex)	24 h post-ischemia induction	Differentiation into astrocytes Neuroprotection Functional outcomes ↑	[60]
Song, M. et al. 2017	Human DPSCs	4 × 10^6^	Rat MCAO (2 h)	Intravenous (tail vein)	24 h post-ischemia induction	Infarct volume ↓ Neurological outcome ↑ Differentiation into astrocytes and neuron-like cells Promoted angiogenesis and inhibited astrocytes	[61]
Kumasaka, A. et al. 2017	Rat DPSCs (dental pulp-derived neurospheres)	1 × 10^6^	Rat severe forebrain ischemia (11 min)	Intravenous (tail vein)	3 h post-ischemia induction	Survival rate ↑ Cognitive functional recovery ↑Reduced the dead neurons of hippocampus CA1	[59]
Nito, C. et al. 2018	Human DPSCs	1 × 10^6^	Rat MCAO (90 min)	Intravenous (tail vein)	Immediately or 3 h post-ischemia	Infarct volume ↓ Neurological outcome ↑ Inflammation modulation	[62]
Sowa, K. et al. 2018	Human DPSCs HGF-transfected DPSCs	1 × 10^6^	Rat MCAO (90 min)	Intravenous (tail vein)	Immediately post-ischemia	Infarct volume ↓ Neurological outcome ↑ Inflammation modulation Promoted angiogenesis	[63]
Zhang, X. et al. 2018	Rat DPSCs	1 × 10^6^	Rat MCAO (2 h)	Intravenous (tail vein)	24 h post-ischemia	Infarct volume ↓ Edema volume ↓ Differentiation into neuron-like cells	[64]

HGF, hepatocyte growth factor; MCAO, middle cerebral artery occlusion. “↑” (means increase) and “↓” (means decrease).

**Table 4 ijms-21-06718-t004:** Experimental study for the neural stem cells (NSCs) transplantation into ischemic brain injury in animal model.

Authors, Year	Cell Type	Experimental Model	Procedure of Transplantation, Timing	Results	Reference
Wang, G. et al. 2020	NSCs (transducted with circHIPK2 siRNA)	Mice MCAO	Intracerebral, 7 days post-ischemia induction	Neural differentiation ↑Neuronal plasticity in the ischemic brain ↑ Long-lasting neuroprotection Functional deficits ↓	[93]
Kondori, B.J. et al. 2020	NSCs isolated from rat SVZ	Rat MCAO	Intra-arterial, 1 day post -ischemia	Infarct size and volume ↓Neurological outcome ↑	[92]
Kim et al. 2020	human neural stem cells (NSCs) encoding gene of choline acetyltransferase (F3.ChAT), an acetylcholine-synthesizing enzyme	Rat MCAO	Intravenous, 2 h post-ischemia	Infarction volume ↓ Cognitive dysfunction ↓Behavioral deficits ↓	[91]
Tian et al. 2019	Leukemia inhibitory factor (LIF)-transfected NSCs	Rat MCAO	Intravenous, 6 h post-ischemia	Infarction volume ↓ Neurological recovery ↑ Glial cell regeneration ↑ White matter injury ↓	[90]
George et al. 2017	Electrically preconditioned hNPCs	Rat MCAO	Intracerebral, 7 days post-ischemia induction	Functional outcomes ↑	[89]
Hou et al. 2017	NSCs	Mice photothromboticischemia stroke model	Intracerebral, 2 days post-ischemia induction	Infarct size and volume ↑ Functional recovery ↓ Neurogenesis ↑	[88]
Zhu et al. 2017	NSCs (Noggin-transfected)	Rat MCAO	Intracerebral, 3 days post-ischemia induction	Neurological scores ↑ Apoptotic neurons ↓ Neuronal morphological damage ↓	[87]
Bacigaluppi et al. 2016	Neural precursor cells	Mice MCAO	Intracerebral, 3 days post-MCAO	Synaptic strength ↑ Functional recovery ↑ VEGF ↑	[86]
Abeysinghe et al. 2015	Pre-differentiation of NSCs into GABAergic neurons	Rat MCAO	Intracerebral, 7 days post-MCAO	Motor function ↑ Proliferation ↑ Neurogenesis ↑	[85]
Yao et al. 2015	Induced NSCs and NSCs	Rat MCAO	Intracerebral, 2 days post-MCAO	Intracerebral lesion size ↓Functional recovery ↑	[84]
Cheng et al. 2015	NSCs	Rat MCAO	Intravenous injection, 1 day post-MCAO	Functional recovery ↑Neurogenesis ↑	[83]
Rosenblum et al. 2015	Brain-derived neurotrophic factor pretreatment of human embryo-derived NSCs	Mice hypoxia-ischemia model	Intra-arterial injection, 3 d post-hypoxia-ischemia	Neuroprotection ↑ Survival ↑ Functional recovery ↑	[82]
Song et al. 2015	Ferumoxide-labeled hNSCs	Rat MCAO	Intravenous injection, 1 day post-MCAO	Infarct volume ↓ Functional recovery ↑Neurogenesis ↑	[81]

MCAO: Middle cerebral artery occlusion. “↑” (means increase) and “↓” (means decrease).

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
