# Peer review of "Recent Advances in Cell-Based Therapies for Ischemic Stroke"

_ijms, 2020, doi:10.3390/ijms21186718_

Round 1

Reviewer 1 Report

Summary: This is a literature review about cell-based therapies for stroke. The authors discuss the use a various cell types, including bone marrow mononuclear cells (MNCs), bone marrow mesenchymal stem (BMSCs), dental pulp stem cells (DPSCs), neural stem cells (NSCs), induced pluripotent stem cells (iPSCs), as well as the genetic modification of stem cells with varying putative beneficial factors. The review discusses the mechanism of action, as far as it is currently known, for each cell type and the promise in transplantation studies conducted to date.

Strength:

  • Well written and well organized. Thorough review of the literature.

Major weaknesses:

  • Figure 1 could be explained better. It is thought that every cell type that is suggested to be used for cell therapy acts via these 2 pathways? For example, iPSC-derived neural progenitors would travel to the spleen? That is what the figure as is may suggest. Or does Figure 1 only apply to MNCs as the cell source? Then it should be re-labeled.
  • More figures would be helpful. For example a figure for the suggested use and putative mechanism with open questions for each cell types may help the reader understand easier, particularly for readers who are visual.
  • For each cell type, it would help to expand a little bit on their definition, where they are harvested from, how they are obtained and how exactly they are used when transplanted. Example line 107-113.
  • What is the reason to have an extensive table with the papers for the DPSCs and NSCs, but not the other cell types? Is it randomly picked?

Minor Weaknesses:

  • It would help a reader who is not directly from the field to reduce the use of abbreviations, there are very many and it makes the reader scroll back to refresh the memory often. Especially in titles.
  • The sentence on line 186 (Recent studies have also shown that engrafted human BMSCs can survive in damaged central nervous system (CNS) tissue and exert immunomodulatory effects by upregulating…) does not seem to fit in that section.
  • Are DPSCs considered MSC? If not the sentence on line 215 also seems out of place: (The optimal route of MSC administration has been investigated, with studies… )

Author Response

We are grateful to the editors and reviewers for their insightful comments and useful suggestions as they have helped us considerably improve our manuscript. As indicated in our responses, we have carefully considered these comments and suggestions during the revision of our manuscript.

Reviewer #1

Major weaknesses:

  1. Figure 1 could be explained better. It is thought that every cell type that is suggested to be used for cell therapy acts via these 2 pathways? For example, iPSC-derived neural progenitors would travel to the spleen? That is what the figure as is may suggest. Or does Figure 1 only apply to MNCs as the cell source? Then it should be re-labeled.

Response: In accordance with the reviewer’s suggestion, we have added some information (administration route and cell type) in figure and figure legends.

  1. More figures would be helpful. For example a figure for the suggested use and putative mechanism with open questions for each cell types may help the reader understand easier, particularly for readers who are visual.

Response: Thank you for your comment. We have added some information about administration route and cell type in figure. Total word count is more than 13,000 and we have already 4 tables and 1 figure. It would be too long.

  1. For each cell type, it would help to expand a little bit on their definition, where they are harvested from, how they are obtained and how exactly they are used when transplanted. Example line 107-113.

Response: In accordance with the reviewer’s suggestion, we have added some sentences as follows.

Line 51: Bone marrow MNCs include a population of monocytes, lymphocytes, mesenchymal and hematopoietic stem cells, and hematopoietic and endothelial progenitor cells [9].

Lines 332–333: Mice-induced pluripotent stem (iPS) cells were first established by Yamanaka et al with introducing of four transcription factors (Oct3/4, Klf4, Sox2, and c-myc) into mouse fibroblasts [117].

  1. What is the reason to have an extensive table with the papers for the DPSCs and NSCs, but not the other cell types? Is it randomly picked?

Response: We agree with the reviewer’s comment, and we have added Tables with MNCs and BMSCs.

Minor Weaknesses:

  1. It would help a reader who is not directly from the field to reduce the use of abbreviations, there are very many and it makes the reader scroll back to refresh the memory often. Especially in titles.

Response: In accordance with the reviewer’s suggestion, we have changed titles.

  1. The sentence on line 186 (Recent studies have also shown that engrafted human BMSCs can survive in damaged central nervous system (CNS) tissue and exert immunomodulatory effects by upregulating…) does not seem to fit in that section.

Response: As pointed out by the reviewer, the line 186 does not fit in that section.

Recent studies have also shown that engrafted human BMSCs can survive…

⇒Recent studies have also shown that engrafted human DPSCs can survive…

  1. Are DPSCs considered MSC? If not the sentence on line 215 also seems out of place: (The optimal route of MSC administration has been investigated, with studies… )

Are DPSCs considered MSC? If not the sentence on line 215 also seems out of place: (The optimal route of MSC administration has been investigated, with studies… )

Response: In accordance with the reviewer’s suggestion, we have changed as follows.

The optimal route of MSC administration has been investigated…

→The optimal route of transplanted stem cells has been investigated…

Reviewer 2 Report

The review on "Recent advances in cell-based therapies for ischemic stroke" is a well written, comprehensive and updated article on the selected topic. It contains a clear and updated summary of all main strategies on the issue of cell-based therapies of ischemic stroke. It will certainly be of great help for readers interested on learning about new strategies on this important issue for the treatment of ischemic stroke patients.

Author Response

Thank you so much for your review and comment of our manuscript.